# PRMT5: An Emerging Target for Pancreatic Adenocarcinoma

**DOI:** 10.3390/cancers13205136

**Published:** 2021-10-13

**Authors:** Michael K. C. Lee, Sean M. Grimmond, Grant A. McArthur, Karen E. Sheppard

**Affiliations:** 1Cancer Research Division, Peter MacCallum Cancer Centre, Melbourne, VIC 3000, Australia; michaelkc.lee@petermac.org (M.K.C.L.); grant.mcarthur@petermac.org (G.A.M.); 2The University of Melbourne Centre for Cancer Research, Parkville, VIC 3010, Australia; Sean.grimmond@unimelb.edu.au; 3Sir Peter MacCallum Department of Oncology, University of Melbourne, Parkville, VIC 3010, Australia; 4Department of Biochemistry and Pharmacology, University of Melbourne, Parkville, VIC 3010, Australia

**Keywords:** PRMT5, protein methyltransferase, pancreatic adenocarcinoma, alternative splicing, FBXW7, c-Myc, GSK3β

## Abstract

**Simple Summary:**

The burden of pancreatic ductal adenocarcinoma (PDAC) increases with rising incidence, yet 5-year overall survival remains poor at 17%. Routine comprehensive genomic profiling of PDAC only finds 2.5% of patients who may benefit and receive matched targeted therapy. Protein arginine methyltransferase 5 (PRMT5) as an anti-cancer target has gained significant interest in recent years and high levels of PRMT5 protein are associated with worse survival outcomes across multiple cancer types. Inhibition of PRMT5 in pre-clinical models can lead to cancer growth inhibition. However, PRMT5 is involved in multiple cellular processes, thus determining its mechanism of action is challenging. While past reviews on PRMT5 have focused on its role in diverse cellular processes and past research studies have focused mainly on haematological malignancies and glioblastoma, this review provides an overview of the possible biological mechanisms of action of PRMT5 inhibition and its potential as a treatment in pancreatic cancer.

**Abstract:**

The overall survival of pancreatic ductal adenocarcinoma (PDAC) remains poor and its incidence is rising. Targetable mutations in PDAC are rare, thus novel therapeutic approaches are needed. Protein arginine methyltransferase 5 (PRMT5) overexpression is associated with worse survival and inhibition of PRMT5 results in decreased cancer growth across multiple cancers, including PDAC. Emerging evidence also suggests that altered RNA processing is a driver in PDAC tumorigenesis and creates a partial dependency on this process. PRMT5 inhibition induces altered splicing and this vulnerability can be exploited as a novel therapeutic approach. Three possible biological pathways underpinning the action of PRMT5 inhibitors are discussed; c-Myc regulation appears central to its action in the PDAC setting. Whilst homozygous MTAP deletion and symmetrical dimethylation levels are associated with increased sensitivity to PRMT5 inhibition, neither measure robustly predicts its growth inhibitory response. The immunomodulatory effect of PRMT5 inhibitors on the tumour microenvironment will also be discussed, based on emerging evidence that PDAC stroma has a significant bearing on disease behaviour and response to therapy. Lastly, with the above caveats in mind, current knowledge gaps and the implications and rationales for PRMT5 inhibitor development in PDAC will be explored.

## 1. Introduction

Since the 1990s, the global pancreatic ductal adenocarcinoma (PDAC) incidences and deaths across 185 countries have increased by 2.3 fold, partly coinciding with the increasing prevalence of diabetes and obesity [1]. Previously difficult to treat cancers like metastatic melanoma and non-small cell lung cancer both have significantly improved 5-year overall survival (OS) with immunotherapy, approximately 50% and 30%, respectively [2,3,4,5,6]. Comparatively, the 5 year OS of all patients diagnosed with PDAC has only marginally improved to 15% over the same period [7]. The poor outcome reflects the fact that less than 20% of all patients diagnosed with PDAC are eligible for upfront surgical resection and complete combination adjuvant chemotherapy [7,8,9,10]. Furthermore, the degree of improvement may be overestimated, given that actual incidences and mortality can be underrepresented in a national registry [11]. This has led to global pancreatic cancer genomic profiling efforts to gain a better understanding of the root causes and potential therapeutic vulnerabilities in this disease. Unfortunately, finding targeted therapies that benefit a reasonable number of subpopulations living with pancreatic cancer remains elusive [12,13,14,15,16,17,18,19].

Nonetheless, genomic profiling has uncovered distinct PDAC molecular subgroups with their own metabolic and prognostic differences that may explain the clinical heterogeneity in PDAC outcomes [20,21,22,23,24,25,26,27,28]. Moffit’s classical versus basal subtype classification appears to be the simplest given its binary nature [24]. However, as the current subtyping methods depend on relative differences in gene expression across a set of genes and a cohort of samples, the issue of relativity limits robust, reproducible classification of all patients into their given molecular subtypes. “Subtype discordant” cases are where the molecular subtype called can be prone to switching depending on the mix of samples in the cohort and display intermediate clinical and molecular characteristics [29]. In reality, single-cell sequencing has revealed that the discordant cases may at least be partially due to intra-tumoral heterogeneity and having variable, co-existing proportions of both classical and basal subtype cells [26]. Incorporation of molecular subtyping into clinical trials has been slow despite the development of single sample classifiers due to the time delays introduced with obtaining RNA sequencing and reproducibility of subtype classification [24,26,30,31]. Nonetheless, treatment implications of these prognostic molecular subtypes are now emerging [30,31]. Moffit’s basal subtype is similar to the better known basal-like subtype in breast carcinoma, though there are no subtype-specific mutations [24]. Moffitt subtypes of PDAC demonstrate enrichment for certain genomic aberrations (e.g., TP53, KTM2A) and altered RNA processing related genes, but the biological significance of the altered transcriptomes has only been partially elucidated [27,32].

Past genomic profiling efforts have also revealed that alteration in RNA processing, with recurrent mutations in splicing factors SF3B1, RBM10 and U2AF1, is one of the hallmark drivers of PDAC tumorigenesis [22]. Unfortunately, this mechanism has not yet been able to be targeted. Recently, Escobar-Hoyos et al. have shown that mutant p53 in PDAC confers an aggressive phenotype via indirectly mediating alternative splicing of GAP17, which results in sustained oncogenic KRAS signalling [33]. At the same time, a mutant p53 PDAC mouse model treated with an SF3B1 (splicing factor) inhibitor (e.g., S3B-8800) showed a more remarkable survival benefit over p53 wildtype PDAC [33]. This suggests that whilst alternative splicing confers selective growth advantage, it also renders PDAC more vulnerable to further perturbation in splicing, similar to MYC-driven tumours [34,35]. These observations support the concept that targeting RNA processing could be a promising strategy in PDAC. However, an alternative splicing factor inhibitor is likely needed, given that SF3B1 is already recurrently mutated in PDAC [22].

Numerous reports demonstrate that high protein arginine methyltransferase 5 (PRMT5) expression is associated with a worse prognosis across multiple cancer types, including PDAC. Inhibition of PRMT5 in preclinical models leads to reduced tumour growth [36,37,38,39,40,41,42,43]. One of the many pleiotropic effects of PRMT5 is the methylation of the small nuclear ribonucleoproteins (snRNP) that govern the site of splicesome activity and a repertoire of protein and RNA interactions. Therefore, unsurprisingly, PRMT5 inhibition results in significant perturbation in alternative splicing [44]. Given the emerging evidence that alternative splicing is both central to PDAC tumorigenesis and renders PDAC susceptible to drugs targeting this process, PRMT5 inhibitors therefore present as a potential novel therapeutic class to combat this malignancy. The main biological pathways reported to be involved in PRMT5 inhibition are p53/MDM4, FBXW7/c-Myc, and AKT/GSK3β/β-Catenin. However, the relationship between cancer growth inhibition and the observed PRMT5 inhibitor effect on altered splicing and these three pathways remains unclear. While c-Myc regulation is likely critical in mediating the deleterious effects of overexpressed PRMT5 in PDAC, the interdependency and the relative importance of different pathways modulating c-Myc activity (FBXW7 epigenetic repression vs GSK3β driven) requires further elucidation. Furthermore, with the increasing recognition of the importance of pancreatic stroma in governing its metastatic potential and the response to therapy, the implications for PRMT5 inhibitors in this context also needs to be considered, given their purported immunomodulatory effects [45,46,47,48,49,50,51,52].

## 2. Clinical Challenges with the Treatment of Pancreatic Ductal Adenocarcinoma (PDAC)

In the best-case scenario, patients newly diagnosed with PDAC undergo both surgical resection and complete six months of modified FOLFIRINOX adjuvant chemotherapy to achieve a median OS of 54 months and 5 year OS of 43% [9,53]. Survival in patients with metastatic disease receiving FOLFIRINOX is much more limited, with an 18 months OS of only 19% [54]. Comprehensive genomic profiling and identification of actionable genomic aberrations only offer limited benefit to 2.5% of patients diagnosed with PDAC due to the rarity of these subgroups and difficulty of access to targeted treatments [13]. Nonetheless, universal germline testing is recommended for all patients diagnosed with PDAC, as 5–20% will have hereditable mutations irrespective of family history [55,56,57]. Pathological germline variants will have hereditary implications but not necessarily personalised treatment implications.

The U.S. Food and Drug Administration (FDA) has approved olaparib for PDAC patients who are carriers of germline BRCA mutations. However, olaparib only confers 3.8 months improvement in progression-free survival (PFS) without any OS benefit, nor improvement in quality of life compared to a truncated course of chemotherapy [58,59]. Although targeting gene fusions involving N-TRK or NRG-1 and using anti-PD1/PDL-1 therapies for microsatellite unstable subtypes have been reported to offer more significant clinical benefits, only limited patients included in the trials/case series had PDAC diagnosis [14,16,18,60,61]. Therefore, some uncertainty remains regarding the true clinical benefit of these therapies for PDAC patients. KRAS mutation is the predominant driver in PDAC but has been largely untargetable until recently in those with G12C mutation, for which the prevalence of mutation is low at 1% [60,62,63]. Although limited mutation targeting therapies are available, they are not universally recommended across international guidelines [61,64,65]. With the increasing incidence of PDAC associated with an ageing population, the poor survival and lack of effective therapies beyond conventional cytotoxic chemotherapy for most PDAC patients, there is a strong clinical need for therapies with a novel mechanism of action [1].

## 3. Protein Arginine Methyltransferase 5 as a Potential Novel Target

Protein arginine methylation is an important post-translational modification that involves protein arginine methyltransferases (PRMTs) catalysing the transfer of methyl group from S-Adenosylmethionine (SAM) to arginine residues on histone and non-histone proteins [66]. The addition of methyl groups makes the protein side-chain more hydrophobic, which alters protein-protein and protein-RNA interactions [67,68]. This can lead to diverse effects on the expression of genes and thus proteins which have been the subject of recent reviews [66,69]. The nine different PRMTs are classified into three types based on their predominant type of arginine methylation. Type I (PRMT1, 2, 3, 4, 5, 6, 8) mediates both asymmetrical dimethylation (ADMA) and monomethylation of arginine (MMA), type II (PRMT5, 9) mediates symmetrical dimethylation (SDMA) and MMA and type III (PRMT7) mediates MMA solely [70]. PRMT5 is the predominant type II methyltransferase that has a breadth of activity in multiple cellular processes [71], including DNA repair [45,46], cell stemness and survival [72,73,74,75], primordial germ cell induction [76,77], ribosome biogenesis [78], RNA transport and Golgi body integrity [79], and regulation of metabolism [80]. In addition, PRMT5 regulates gene expression through multiple mechanisms, including histone methylation mediated transcriptional suppression [81], alteration of protein interactions [82] and use of alternative gene promoters [83]. PRMT5 has also been shown to play a vital role in splicesome fidelity and activity, altering the predominant transcript/isoform used [84] and through post-translational modification of proteins, reducing protein stability [85] and signal transduction [86] (Figure 1).

During the past 10 years, PRMT5 has emerged as a potential therapeutic target in multiple solid tumours [36,37,38,39,40,41,42] and haematological malignancies [87]. High PRMT5 expression in PDAC patients is associated with 5 year OS of 13% versus 46% in low expressors [88]. PRMT5 knockdown via short hairpin RNA or utilising an inhibitory compound like GSK3326595 is associated with reduced proliferation in pancreatic cell lines and organoids and reduced FDG-PET avidity in xenograft models of PDAC [85,89]. There is also evidence that PRMT5 inhibition increases sensitivity to gemcitabine, a commonly used chemotherapeutic agent in PDAC [90,91]. It has also emerged that KRAS mutation correlates with higher PRMT5 expression in colorectal cancer and increased susceptibility to PRMT5 inhibition [92]. If a similar predisposition is seen in PDAC, this could have profound significance given that >90% of PDAC have a KRAS mutation [20].

The effect of PRMT5 inhibition on PDAC growth is also intriguing as this may be mediated by its effect on alternative splicing and may represent a novel therapeutic class [93]. Aberrant splicing has been recognised as a mechanism that promotes PDAC tumorigenesis and therapeutic resistance [94,95,96,97,98,99,100,101]. Emerging evidence suggests that alternative splicing dependent tumours, like mutant p53 PDAC and both MYC and SF3B1 (splicing factor) mutation-driven cancers, are more vulnerable to further perturbation in splicing [33,34,95,102,103,104,105,106,107]. Therefore, PRMT5 inhibition presents an attractive therapeutic strategy to perturb splicing via methylation-dependent modulation of spliceosome assembly and inducing genotoxicity and cell death via epigenetic reprogramming [74,108,109]. This has led to phase I trials of PRMT5 inhibitors (GSK3326595 and JNJ64619178) in haematological malignancies and solid cancers. Their respective activities in head and neck squamous cell carcinoma, breast and adenoid cystadenocarcinoma have recently been reported with proven safety [110,111]. Similar trials with PF-06939999 (NCT03854227), PRT811 (NCT04089449) and PRT543 (NCT03886831) are currently ongoing. While PRMT5 inhibitors have not yet been trialled in PDAC, pre-clinical support for its inclusion in trials is accumulating.

The translation of PRMT5 into a therapeutic target has been slow due to challenges in identifying the relevant mechanism of action, since PRMT5 is involved in a plethora of pathways and multilevel control of gene and protein expression (Figure 1) [112]. Furthermore, PRMT5’s control of gene expression can be modulated by other methyltransferases. For example, PRMT7 methylates distal H4R17 arginine residues on histone tail, which enhances PRMT5 methylating efficiency on histone H4R3 in a process called positive cooperativity [113]. This increased efficiency of post-translational methylation by PRMT5 may alter the rate and scope of biological response. However, the synergy generated by positive cooperativity may go undetected by conventional differential gene expression analysis if mRNA expression remains unchanged. PRMT5 can act as an epigenetic repressor for micro RNA (miRNA) expression to enhance c-Myc and cyclin D1 signalling [114]. Furthermore, PRMT5 itself can also be regulated by miRNA and long non-coding RNA [115]. The functional significance of these non-genetic mutation factors is generally less understood. Therefore, it conceptually adds to the complexity when deciphering the underlying mechanistic process that underpins PRMT5 inhibitors’ efficacy.

## 4. Decoding the Mechanisms of Action of PRMT5 Inhibitor in Cancers

Although PRMT5 has differing roles in many cellular processes, three characterised pathways are thought to mediate the growth inhibition effect seen with PRMT5 inhibitors. Depending on which biological pathway is pertinent to the growth inhibitory action of PRMT5, different PDAC molecular subtypes may have different responses to PRMT5 inhibitors.

### 4.1. p53/MDM4 Axis

One of the best-characterised effects of PRMT5 inhibition is the upregulation of p53 activity through alternative splicing of MDM4 (Figure 2). PRMT5 inhibition promotes skipping exon 6 of MDM4 (shorter, less stable MDM4 isoform), resulting in the loss of the p53 interacting domain and thus releasing MDM4′s repression of p53. In p53 wildtype cells, this leads to increased p21 expression and RB mediated G1 cell cycle arrest and cell death [91,116,117]. Regulation of MDM4 by PRMT5 inhibition has been demonstrated in p53 wildtype melanoma, breast, lymphoma and multiple myeloma cell lines and mouse models [91,116,117]. The introduction of exogenous full-length MDM4 reduces sensitivity to PRMT5 inhibition, with reduced p21 activation but no change in p53 protein levels. Therefore, the induction in p53 activity may be due to other mechanisms such as direct methylation by PRMT5 [118]. In mutant p53 cell lines, no change in p21 levels were observed with PRMT5 inhibition [82]. Thus, the relevance of this axis to PDAC is less certain when considering 65–85% of PDAC have mutant p53 [21,119,120].

### 4.2. AKT/GSK3β/β-Catenin Axis

In PDAC cell lines, PRMT5 indirectly induces autophosphorylation of tyrosine residues Y1068 and Y1172 on epidermal growth factor receptor (EGFR) (Figure 2). It is associated with increased phosphorylation and activation of AKT and subsequent phosphorylation mediated inhibition of GSK3β and activation of β-Catenin, which enhances epithelial to mesenchymal transition and increased metastatic potential (Figure 2) [108]. Similar dysregulation of this axis has been observed in colorectal and lung cancer [109,121]. Of interest, PRMT5 methylation of the EGFR receptor at Y1175 positively modulates the phosphorylation of EGFR Y1173 residue in the breast cancer setting. However, it has the opposite effect on cellular growth to those seen in PDAC, with attenuation of ERK signalling [86]. Whilst reduction in PTEN activity through promoter suppression by PRMT5 can lead to an alternative mechanism of upregulation of this axis in glioblastoma, the same is not observed in PDAC, lung cancer and gastric cancer [42,90,122,123]. In lymphoma cell lines, PRMT5 mediates epigenetic repression of AXIN2 and WIF1, leading to indirect activation of AKT, which could be another alternative upstream mechanism; however, this has not been investigated in PDAC [124]. This array of modulators to AKT/GSK3β/β-Catenin highlights that the mechanism of action of PRMT5 will have to be validated in each cancer type, whilst the repeated alterations within this axis argue for its significance.

If the PRMT5 inhibitory effect is mediated via the GSK3β axis, its effectiveness may be limited to a subset of PDAC based on the effects seen with direct GSK3β inhibitor. Despite KRAS mutation being the main driver for PDAC tumourigenesis, only a subset of KRAS mutant shows dependency where siRNA inhibition of KRAS induces apoptosis and caspase-mediated death. Similarly, GSK3β inhibition selectively causes apoptosis in these “KRAS dependent” PDAC [125]. More recently, direct GSK3β inhibition has been shown to suppress growth in Moffitt’s basal molecular subtype PDAC, but not in the classical subtype [30]. These observations suggest that PRMT5’s clinical efficacy may be dictated by the molecular background seen in the cell line models [125]. Whilst GSK3β may be an essential node of control, it has multiple downstream effectors, including NFkβ and mTOR pathways [126]. The significance of these parallel pathways in mediating the growth inhibitory effect of PRMT5 inhibitors has not been considered.

The ability of PRMT5 to regulate pathways at multiple levels further adds to the complexity in deciphering its primary mechanism of action. For example, PRMT5 can modulate the NFkβ pathway via (1) regulation of GSK3β and a subsequent inhibitory effect on IKK, (2) direct methylation of p65 subunit l, preventing translocation of NFkβ into the nucleus [127] (Figure 2), and lastly (3) repression of FBXW7 leading to increased NFKβ2 expression [128]. The multiple points of control on biological pathways seem to be a recurring theme and pose challenges to the mechanistic understanding of PRMT5. It raises the question of whether targeting multiple points in a pathway is necessary for its growth inhibitory effect or if this redundancy may confer an advantage where resistance to therapy is less likely to emerge.

### 4.3. FBXW7/c-Myc Axis

FBXW7 is a potent tumour suppressor gene that functions as part of the Skp1-Cullin1-F-box (SCF) ubiquitin ligase complex that targets major oncoproteins, including Notch-1 [116], c-Myc, cyclin E [117], c-Jun, and E2F1 for ubiquitination and subsequent degradation [119]. Unsurprisingly, recurrent deletion and mutation in FBXW7 is seen in many cancer types and is associated with a worse prognosis [120,122,123,129,130]. In PDAC, FBXW7 expression is commonly downregulated due to ERK phosphorylation of FBXW7, leading to its polyubiquitination mediated degradation, with subsequent stabilisation of c-Myc [131,132,133]. In patients with high PRMT5 expression, FBXW7 expression is further lowered due to methylation mediated epigenetic silencing of the FBXW7 promoter (Figure 2) [85]. Conversely, PRMT5 inhibition in PDAC xenografts upregulates FBXW7 expression with a corresponding reduction in c-Myc protein stability and glycolytic activity [85].

Of note, the degradation of MYC requires priming by ERK-dependent phosphorylation at serine 62 followed by GSK3B mediated phosphorylation at threonine 58 before MYC can interact with FBXW7 for polyubiquitination mediated degradation [119]. Therefore, it is unclear if downregulation of c-Myc induced by PRMT5 inhibition is predominantly due to increased GSK3B mediated phosphorylation of c-Myc or upregulation of FBXW7 gene expression, or via the indirect effect of both axes on reducing ß-Catenin (Figure 2) [134]. To further complicate PRMT5’s role in c-Myc expression, PRMT5 also has multiple other pleiotropic effects on both the MAPK and PI3K pathways that can modulate MYC activity (Figure 2) (see [135] and review [136]). The relative importance of these other pathways has yet to be assessed in the PDAC context and whether these accessory pathways lead to therapy resistance or synergy is uncertain. Adopting an unbiased systematic approach, coupled with analysis of multi-omic changes over different time points would assist in identifying critical essential genes and decipher the order of activation that may govern the temporal activity of PRMT5 and the diversity of interaction.

## 5. PRMT5 and Alternative Splicing: Potential Key Mechanism of Action of PRMT5 Inhibitors

Besides identifying the alterations in signalling in different biological pathways, examining how alternative splicing is perturbed may shed light on PRMT5 inhibitors’ mechanism of action as the two are seemingly linked. PRMT5 inhibition leads to thousands of alternative splicing events (ASEs) with enrichment in genes in RNA processing [93]. However, it is challenging to discern which of the thousands of ASEs are pathways of interest with confidence or make any claims beyond prognostic associations or pathway enrichment. Nevertheless, baseline differences in the top 45 differential splicing events in glioblastoma can form a splicing signature that seems to differentiate between good and poor responders to PRMT5 inhibition better than mutational status [137]. Similarly, greater activation of caspases and subsequent apoptosis is observed following PRMT5 inhibition in cells with splicing factor gene mutations (i.e., SF3B1) [37,113,138]. These observations further support the need to account for baseline differences in altered splicing in PDAC when interpreting the response to PRMT5 inhibitors, especially when certain spliced variants are themselves associated with prognostic implications [139,140].

Curiously, the addition of PRMT1 to PRMT5 inhibition in a pancreatic cancer cell line induces a synergistic increase in >2400 novel ASEs compared with PRMT5 or PRMT1 inhibitor alone. Conversely, only approximately 220 ASEs were commonly induced by both inhibitors individually [93]. Dual combination treatment ameliorates the compensatory increase in methylation by other PRMTs when PRMT1 is inhibited alone, resulting in more complete inhibition of all types of methylation [93,131]. It is unclear whether the synergy in cancer growth inhibition from the combination of PRMT inhibitors arises from more complete inhibition/methylation of common substrates or whether synergy arises from apparent functional redundancy. There are examples where both PRMTs similarly regulate a common substrate, such as SPT5, modifying its interaction with RNA polymerase II [141]. However, the two PRMTs can also have an opposite effect on common substrates such as p65 in the NFkB pathway [127,142] and the regulation of E2F1 in determining cell survival [143,144,145]. Detailed examination of the individual and common ASEs associated with PRMT1 and PRMT5 inhibition has not been reported, nor has the impact of the novel repertoire of ASEs associated with inhibiting the PRMTs in combination. Despite the synergy, little is known regarding the role of PRMT1 in alternative splicing other than in acute megakaryoblastic leukaemia, where PRMT1 alters the splicing activity of a splicing factor, RBM15 and blocks cellular differentiation [132].

Similarly, in acute myeloid leukaemia (AML) cell lines, combination PRMT1 and PRMT5 inhibition also result in synergistic increases in novel ASEs [133]. The vast majority of the altered splicing results in changes to the canonical splicing pattern and only <20% are novel splicing events [133]. Whilst non-canonical changes may seem subtle, splicing changes with RBM10 and the MDM4/p53 axis mentioned above can significantly influence tumorigenesis [118,146]. Likewise, altered splicing of splicing factor SRSF1 can alter the repertoire and affinity of the mRNAs it interacts with, for which the biological consequences remain only partially elucidated [133,147]. In AML, the enhanced growth inhibition with combination treatment was partially mediated by downregulation of essential survival genes and increased EZH2 expression via alternative splicing by SRSF2 [35,133]. Whilst re-expression of EZH2 has a tumour suppressor effect in myeloid malignancies, it is the depletion of EZH2 in PDAC that is associated with conversion towards the less aggressive, classical molecular subtype [31,148]. Therefore, mechanisms suggested in other cancers may not translate to PDAC, and thus further validation is warranted.

Reliance on published gene expression data and splicing analysis following PRMT5 treatment at a single timepoint limits the potential to assess the biological effects of PRMT5 inhibitors comprehensively. Detained introns from alternative splicing impair mRNA transport out of the nucleus, thus reducing mRNA translation (Step 6, Figure 2) [138,149]. PRMT5 also methylates various heterozygous nuclear ribonucleoproteins (hnRNPs) that directly modulate mRNA export out of the nucleus [138]. Positive cooperativity of PRMT5 with other methyltransferases and the aforementioned mechanisms do not necessarily change mRNA expression level, but may still alter protein expression significantly. Based on differential gene expression analysis, the mechanistic effect of PRMT5 inhibition in glioblastoma may be related to PRMT5 induced altered expression of genes associated with cell regulation [137,147]. However, a wide variation in growth inhibition is seen across patient-derived samples, which was better explained by pre-existing splicing differences between molecular subtypes [137]. This again supports the importance of alternative splicing as a mechanism of action for PRMT5 inhibition in a way previously not considered. The spatial and temporal control of PRMT5 by its co-binding factors likely contributes towards the time, spatial and dose-dependent effects of PRMT5 inhibitors [84]. Thus, the current single time point RNASeq may not fully describe the effects of PRMT5 inhibitors.

## 6. Potential Selective Predictive Response Biomarkers

With the pleiotropic effects of PRMT5 and the multitude of possible mechanisms of action of PRMT5 inhibitors, a predictive biomarker is ever more critical to better select appropriate patients who will benefit from targeted PRMT5 inhibition. Unfortunately, currently proposed biomarkers of sensitivity to PRMT5 inhibition are inadequate at predicting the growth inhibitory effect and thus preclude their incorporation into clinical practice even though they offer mechanistic insights.

The most commonly cited biomarker of sensitivity is the homozygous deletion of the methylthioadenosine phosphorylase (MTAP) gene. This occurs in 25% of PDAC, owing to the MTAP gene’s proximity to the frequently deleted *CDKN2A* gene on chromosome 9 [150,151]. Biologically, MTAP is involved in the metabolism of 2-methylthioadenosine (MTA) and regenerates methionine needed for S-adenosyl-L-methionine (SAM) production. In the absence of MTAP, accumulated MTA acts as an intrinsic and selectively competitive inhibitor with a >100 fold preference for PRMT5 SAM binding pocket over other PRMTs. This reduces SAM binding to PRMT5 and in turn, limits PRMT5’s methyltransferase activity [152,153,154,155,156]. In MTAP deleted pancreatic organoids, increased PRMT5 inhibitor sensitivity with more significant growth inhibition were observed, whilst resistance was seen in the MTAP wildtype organoids [89]. Of note, a subset of MTAP wildtype organoids demonstrated similar sensitivity to PRMT5 inhibition as MTAP deleted organoids, where the degree of sensitivity was better predicted by MTA level rather than MTAP status [89]. This may reflect that the binding of MTA to PRMT5 also impairs the binding of GSK3326595 to the substrate-binding pocket [154]. Therefore, although low MTA and PRMT5 inhibitor concentration showed synergistic enhanced PRMT5 inhibition, the synergy is markedly attenuated at a high concentration of either MTA or PRMT5 inhibitor [50]. Newer generations of PRMT5 inhibitors are designed to escape this competition, so it remains to be seen if MTAP status may still be of relevance [157,158].

Although the relevance of MTAP appears somewhat promising based on pancreatic organoid data alone, examination across a wider number of cell lines suggests MTAP status predictiveness is less robust. Further examination across 11 pancreatic cell lines and a broader 64 cell line panel revealed no correlation between response to PRMT5 inhibition and MTAP status [152,156]. Equally, there was no difference between SDMA levels (a functional measure of PRMT5 activity) as determined by MTAP status [152]. Nonetheless, in MTAP deleted cell lines, the growth inhibition seen with PRMT5 inhibitor treatment can be recapitulated by inhibiting PRMT5’s co-binding factors (e.g., RIOK1, PICln, MEP50, CORP5) and MATA2A, the enzyme responsible for the production of SAM methyl donor [152,159]. Each co-binding factor directs PRMT5 towards a different cellular process. Yet, all these different cellular processes are seemingly of equal importance to maintaining cellular growth in MTAP deleted cells when the reverse is not seen in MTAP wildtype. Unfortunately, a boutique CRISPR screen of >1000 gene regulators has failed to explain this phenomenon [153]. Therefore, a more detailed assessment of the biological effects of each of the binding partners is needed, or a wider whole-genome CRISPR screen would provide an unbiased approach. Although homozygous deletion of MTAP is associated with increased PRMT5 inhibitor sensitivity in some pancreatic organoids, broader examination of the impact of MTAP status across different tumour cells lines suggests that it is not a reliable biomarker of sensitivity to PRMT5 inhibition [96,160,161]. Nonetheless, MTAP status is a contributory factor representing incompletely characterised molecular differences that partially predict the response to PRMT5 inhibition.

The other common “biomarker” used in the field is measuring the SDMA level, which is a functional measure of PRMT5 specific methylation activity rather than its clinical/biological effect [152,162]. In the METEOR-1 phase one study, despite near-complete suppression of SDMA in both plasma and tissue biopsy after 14 days of treatment with GSK3266595, wide variation from a two-fold increase in size to partial response was seen across several different tumours [110]. Nonetheless, more complete inhibition of SDMA (>80–90%) across 240 cell lines trended towards increased PRMT5 inhibitor sensitivity (growth inhibition) [84]. A similar proxy is H4R3me2, which measures the repressive histone methylation activity of PRMT5 but does not capture the activity of PRMT5 on non-histone proteins [112,156]. A higher ratio of CLNS1A expression to RIOK1 expression had also been proposed as another biomarker [147]. However, similar to all functional biomarkers aforementioned, their relevance outside of the specific molecular context to which they were initially identified is questionable. The disconnect between the enzymatic inhibition of PRMT5 as a target and the observed biological effect is puzzling but probably not surprising given the multitude of effects it has across gene expression and cellular processes.

What is less discussed outside of the haematological context is the relevance of recurrent mutation in splicing factors that may confer increased sensitivity to PRMT5 inhibition, arising from their increased susceptibility to further perturbation of altered splicing [96,163,164,165,166]. For instance, the recurrent mutation in splicing factors such as SRSF2, U2AF1, ZRSR2, or SF3B1 in myelodysplastic syndrome is associated with a more significant reduction in cancer cell numbers in response to PRMT5 inhibition [164,167]. Similar vulnerabilities may exist in PDAC as recurrent mutations in U2AF1, SF3B1 and RBM10 are known drivers in PDAC tumorigenesis [22]. However, whether it is the added global splicing stress or a specific altered splicing event that underpins this synergistic effect remains to be revealed. Some preliminary evidence supports the former, where myelodysplastic syndrome with splicing factor(s) mutations shows excessive R loop formation, leading to DNA damage and subsequent cell death [167]. The challenge will be developing methods to comprehensively measure the change in the repertoire of mRNA and protein interactions due to PRMT5 inhibition and determine how this affects the translational capacity and efficiency, with and without splicing factor mutations [133,160]. Unfortunately, the inability to identify specific subgroups of patients that may benefit from treatment with a PRMT5 inhibitor has contributed to its limited utility and efficacy seen in the current early phase trials [110,111].

## 7. PRMT5 Inhibition and Its Implications for Tumour Microenvironment

With limited targetable mutations in PDAC, attention had turned to target the characteristic desmoplastic stroma that supports PDAC tumorigenesis. Traditionally, the desmoplastic stroma was thought to confer chemotherapy resistance which could be reversed with sonic hedgehog inhibitors in pre-clinical models. However, a clinical trial of this inhibitor led to the paradoxical acceleration of PDAC growth [168]. The sonic hedgehog inhibitor happened to inhibit myofibroblast cancer-associated fibroblasts (CAFs) and also reduced fibrotic type I collagen deposition, leading to loss of its tumour restraining properties [72,169]. The resultant influx of regulatory T cells was also thought to contribute to reduced survival [169]. The spatial arrangement of myofibroblast CAFs near PDAC cells and the more immunosuppressive, inflammatory CAFs in the periphery is maintained by PDAC secretion of TGFB and IL1, respectively [170]. Besides the fibrotic reaction, the pancreatic stroma is characterised by high interstitial pressure due to the presence of hyaluronic acid, which causes vascular collapse and impedes chemotherapy delivery, thus conferring resistance [161]. Despite promising earlier phase trials, pegvorhyaluronidase alfa (targeting hyaluronan) ended in a negative phase III trial [161,171].

Further evidence of PDAC and stroma interaction comes from the observation that mutant p53 PDAC can educate CAFs. These “educated” CAFs can instruct the more indolent wildtype p53 PDAC to adopt a more invasive phenotype, either directly via facilitation of invasion, or indirectly by increasing perlecan/HSPG2 secretion. [35,171]. In the classical molecular subtype of PDAC, the stroma upregulates cholesterol synthesis to support aberrant cholesterol uptake, whilst the basal subtype adopts more of a fibrotic phenotype [172]. These observations highlight the importance of stroma in governing the behaviour of PDAC and supporting the maintenance of the molecular subtype. Perturbation of stroma can have inadvertent consequences on its tumour restraining properties. Thus, the effect of novel PDAC therapies on stroma should be evaluated to mitigate the risk of failure in the clinic.

PRMT5 inhibition may enhance immunogenicity in PDAC, based on the predicted impact on the PDAC microenvironment and effects reported in other cancers. Outside of the microsatellite unstable setting, immunotherapy trials in PDAC have all been negative [173,174], except the COMBAT trial, a phase IIA trial, where there was evidence of activity in combination with a CXCR4 antagonist [175]. This likely reflects the immunosuppressive environment, partly mediated by inflammatory CAFs that arise from IL1a secreted by PDAC and the result of NFKB pathway activation [170]. PRMT5 inhibition may suppress inflammatory CAFs by hypomethylation of p65, which precludes its translocation into the nucleus, thus inhibiting NFKB mediated transcription [127]. In addition, PRMT5 inhibition can also cause alternative splicing of TIP60 (histone acetyltransferase) and decreased methylation of TIP60 cofactor, RUVBL1, both of which lead to decreased TIP60 mediated acetylation of FOXP3 (Figure 2, bottom panel) [45,46]. This results in increased degradation of FOXP3+ regulatory T cells followed by a secondary influx of CD8^+^ T cells [47]. In addition, both enzymes contribute to the mobilisation of 53BP1 and promoting homologous recombination repair and reactivation of p53 mediated apoptosis [82]. In melanoma, PRMT5 inhibition has been shown to increase both neoantigen expression via perturbation of alternative splicing and also immunogenicity with increased MHC class I expression through hypomethylation of NLRC5 [48,49]. These observed effects of PRMT5 may contribute to increased immunogenicity, but they require validation in PDAC.

In contrast, in human peripheral blood mononuclear cells, PRMT5 inhibition reduces CD8+ T cell proliferation via the MDM4/p53 axis and reduces metabolic activity (glycolysis and fatty acid oxidation) via inhibition of AKT/mTOR pathways [50]. PRMT5 inhibition can also impair Th17 cell differentiation, with reduced T helper cell proliferation and cytotoxic lymphocyte recruitment through reduced γc-cytokine signalling [52,176]. It would appear that the proliferation and survival of B cells and T cells are also dependent on PRMT5-facilitated splicing of IL2 receptor and JAK3 [52,177]. Although PRMT5 inhibition appears to reduce activation and recruitment of the immune system in the periphery compartment, PRMT5 inhibition has an immune promoting effect in the tumour microenvironment. The overall net effect of PRMT5 inhibition on the immune system is unclear. There is currently a relative paucity of research in this domain, except for the METEOR-1 study containing an arm combining GSK3326595 with pembrolizumab (NCT02783300).

## 8. Discussion

In the era of precision oncology, melanoma and lung cancer have recently seen giant strides in improving survival outcomes, with the advent of immunotherapy and the identification of subgroups amenable to targeted therapies. Advanced PDAC continues to have a poor prognosis with a median survival of fewer than 12 months [54] and with rising incidences. Therefore, a strong unmet clinical need exists for novel therapies, especially those of a novel therapeutic class. The interest in developing PRMT5 inhibitors stems from the observation across multiple tumour types that high expression of PRMT5 is associated with worse prognosis and its inhibition in various preclinical models has led to PDAC regression. In addition, PRMT5 inhibition mediated alternative splicing could pose as a novel therapy against a hallmark driver of PDAC tumorigenesis that has not been successfully targeted to date [22]. Based on current evidence, there is a possibility that the therapy will only benefit up to 20% of patients who are basal molecular subtype [30]. Regardless, this targeted treatment will still be the only treatment that encapsulates a substantial subgroup within the PDAC cohort [24]. Furthermore, early phase trials have already demonstrated the safety of this class of drugs [110,111]. Thus, PRMT5 is poised for rapid translation into PDAC treatment once the underlying mechanism is better elucidated.

The scientific challenge is that PRMT5 has multiple cellular functions and regulates signalling at multiple levels within a pathway and via differing mechanisms. Nonetheless, the regulation of c-Myc expression thus far seems to be central to mediating the inhibition of cancer growth. The apparent redundancy of inhibitory actions that PRMT5 has on c-Myc stability is not well understood. However, PRMT5 inhibition may result in a longer duration of disease control due to concurrent inhibition of multiple parallel signalling pathways. This potentially reduces the rate of adaptive resistance development that is seen with other targeted therapies. Lack of a singular mechanism of action, as seen with oncogenes, and the varied mechanisms across different cancers suggest that its mechanism of action is likely unique to a particular cancer type. However, given that the response to PRMT5 inhibitor therapy is partially dependent on molecular contexts such as MTAP deletion or altered splicing as seen in glioblastoma, its mechanism may not be tissue-specific but molecularly defined. Therefore, understanding the alternative splicing landscape in PDAC is likely to be important in defining the use of PRMT5 inhibitors for this disease. With the poor response rate seen in early phase trials, there is a strong need for a better predictive response biomarker, as the current “biomarker” of SDMA levels are functional measures of PRMT5 activity but not necessarily representative of its biological effect [152].

To advance the translation of PRMT5 inhibitors to a clinical trial in PDAC, better elucidation of the PRMT5 inhibitor’s mechanism of action and/or identification of appropriate predictive response biomarkers will be paramount. Firstly, acknowledging the plethora of potential actions of PRMT5, unbiased and systematic assessment of which genes PRMT5 inhibition affects will be beneficial to narrow down biological pathways of relevance. Examples of these approaches would include whole-genome CRISPR screens to identify the essential genes required for response to PRMT5 inhibition and transcriptomic assessment of differential gene expression and splicing changes. However, reliance on conventional knock-out screens or transcriptomic analysis alone is insufficient to elucidate the multiplicity of effects that PRMT5 can have over gene and protein expression. Therefore, multi-omic strategies, including proteomic, are likely needed to comprehensively ascertain the relative importance and interdependency of different mechanisms that PRMT5 can concurrently modulate, such as epigenetic, translational efficiency, RNA transport, protein stability and crosstalk with phosphorylation and ubiquitination processes [178] (Figure 1). Nonetheless, the benefit of being in the precision oncology era is the accessibility of both bioinformatics and laboratory-based approaches to tackle the complexity of PRMT5 as a biological target and potentially shorten the drug target’s translational timeline.

Lastly, with the emerging evidence suggesting the importance of stroma in determining PDAC behaviour and the dynamic nature of the stroma, PRMT5 inhibitor development in PDAC must be considered in this context. It remains to be further elucidated whether the pro-immunomodulatory effect PRMT5 inhibition has on the microenvironment is sufficient to override the inhibition of immune cells in the peripheral system. This is particularly important in the PDAC setting, where there are insufficient numbers of activated immune cells to orchestrate an immune response, perhaps explaining the negative findings in immunotherapy trials to date.

With all these caveats, PRMT5 remains a potential target for PDAC, and its ability to modulate alternative splicing provides a novel therapeutic class for the treatment of this disease. Biologically, with PRMT5 having multiple control points within and across differing signalling pathways, this agent may be less likely to induce tumour adaptation and resistance, as seen in many targeted therapies [179]. Currently, trials combining PRMT5 inhibitors with 5-azacitidine (NCT03614728) and Pembrolizumab (NCT02783300) are planned. Other rational combinations may be possible with an improved understanding of its underlying mechanism of action. PRMT5 as a target has its merit, but overall, further research is needed to maximize its utilisation in PDAC cancer care.

## 9. Conclusions

In view of limited targeted therapy options available and continued poor survival with conventional cytotoxic chemotherapy, novel therapeutic options are needed to improve the survival outcomes of patients with advanced PDAC significantly. PRMT5 inhibition poses an interesting target given its effects on alternative splicing, a known hallmark driver in PDAC pathogenesis. The pleiotropic effects of PRMT5 inhibition may depend on the molecular context and thus differ from conventional oncogene inhibitor therapy. While this poses challenges to understanding its mechanism of action and appropriate patient selection, PRMT5 inhibition may result in less chance of therapy resistance, due to redundancy in its actions on either the same or complimentary signaling pathways. Therefore, PRMT5 inhibition is a note worthy target to further develop for the treatment of PDAC.

## Figures and Tables

**Figure 1 cancers-13-05136-f001:**
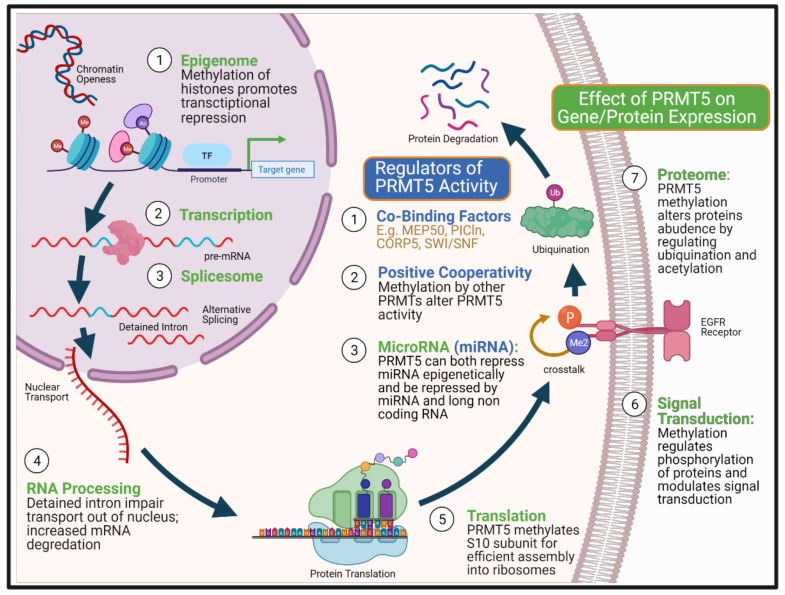
Multi-level control of gene and protein expression by PRMT5. The figure summarizes biological points at which PRMT5 can impact on both gene and protein expression (green colour), from epigenetic repression to post-translational modification of the protein, altering both the stability and capacity for signalling. PRMT5 achieves this complex multi-level of control via methylation of arginine residues on histones and non-histone proteins. PRMT5’s enzymatic activity is also modulated by three types of factors (blue colour). First, by PRMT5’s co-binding factors that determine its substrate fidelity as well as the location of action. Part of PRMT5 pathogenicity arises from its suppression of less well characterised non-coding RNA as well as being regulated by these microRNAs. Its positive cooperativity with other PRMTs may alter the rate and scope of biological response from protein interactions without necessarily causing changes in mRNA transcript expression. The complexity of PRMT5 interactions therefore hinder its development as a therapeutic target. Created with BioRender.com.

**Figure 2 cancers-13-05136-f002:**
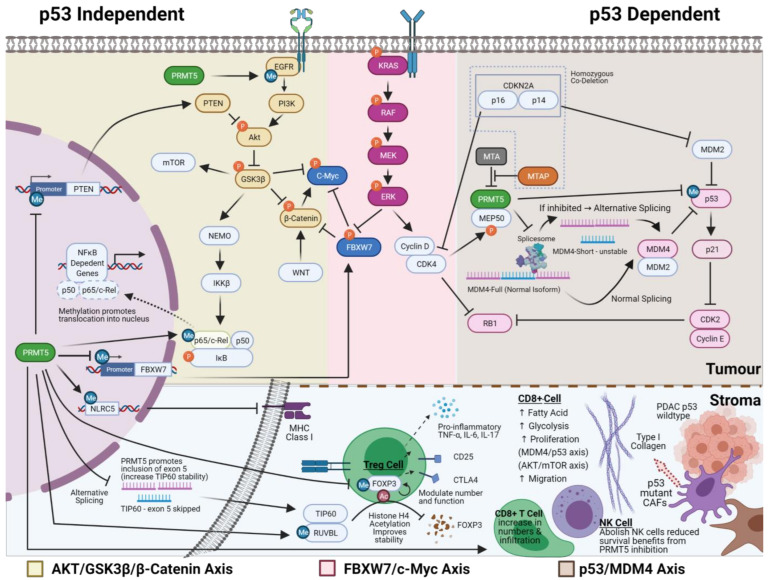
The mechanism of action of PRMT5. The figure portrays three best characterised axes of how PRMT5 can drive cancer growth that may be of relevance in PDAC, namely AKT/GSK3-β/β-Catenin axis, FBXW7/c-Myc axis and p53/MDM4 axis. The complex interplay between methylation, phosphorylation, acetylation and ubiquitination are denoted by blue (Me), orange (P) and brown (Ac) colours, respectively. The bottom panel highlights some of the possible effects that PRMT5 inhibition may have on the tumour microenvironment and how this relates to what we know about PDAC stroma and its interplay with PDAC tumour cells. Created with BioRender.com.

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
