# Peer review of "PRMT5: An Emerging Target for Pancreatic Adenocarcinoma"

_cancers, 2021, doi:10.3390/cancers13205136_

Round 1

Reviewer 1 Report

The manuscript entitled „PRMT5: Emerging Target for Pancreatic Adenocarcinoma.“ from Michael K. C. Lee et al. summarizes current knowledge and investigations on the role of PRMT5 in pancreatic adenocarcinoma (PDAC). The authors provided a brief overview on current therapeutic strategies and of molecular subgroups of PDAC. Moreover, they described pathways affected by PRMT5 and possible therapeutic approaches exploiting PRMT5´s molecular mechanism of action, as well as potential PRMT5-dependent biomarkers and the effect of PRMT5 on the tumor microenvironment.

Overall, the manuscript is well written and includes figures that display a good overview on the topic. There are a few issues, which should be addressed before publication:

  1. Page 2, line 60: Please rephrase the first sentence describing a “borderline case”. Anyway, I find the name “borderline” problematic, since we classify the tumors in resectable and borderline-resectable cases, which could lead to misunderstandings with molecular “borderline” subtypes.
  2. Page 2, line 68: Please be more precise which type of cancer you mean having an “enrichment of certain genomic aberrations”.
  3. Page 3, line 110: Please remove the word “remaining”.
  4. Beginning of chapter 2: The details on the overall survival times are confusing. Could you please state either median OS in month or 5-year survival rate in percent.
  5. Page 3, line 129: As far as I know, targeted therapies are available only for patients in clinical trials. There is no targeted therapy recommendation in any treatment guideline. This has to be pointed out. Moreover, the sentence that “alternative strategies such as targeting the PDAC desmoplastic stroma have thus so far been unsuccessful” does not fit in this context.
  6. On page 3, you stated that PRMT5 is a type II PRMT. In the cited paper only PRMT1, PRMT2, PRMT3, PRMT4, PRMT6 and PRMT8 are described to dimethylate asymmetrically and that PRMT1 and PRMT5 may monomethylate arginine, which however needs further investigations. Please check the literature and your statement in the manuscript.
  7. Page 4, figure 1: Is there a specific reason for the brown coloration of the names of the co-binding factors? Does it refer to the color of the arrow from Me2 to P at EGFR?
  8. On page 6: “the clinical response may be restricted to the 20% of PDAC belonging to the basal molecular subtype. This is on the basis that direct GSK3beta inhibition has recently been shown to suppress growth in Moffitt’s basal molecular subtype PDAC only and not in the classical subtype.” - Please provide a reference.
  9. Please cite Figure 2 in the main text when addressing the PRMT5-regulated pathways that are shown in the figure.
  10. On page 9: “… impact of the MTAP status across different histologies suggests …” - What kind of histologies? Different tumor entities?
  11. Some of the references are missing in the discussion (e.g. Page 12, line 525, line 526). Please include them, although you might just repeat some previous statements from the other chapters.
  12. It is quite obvious that chapter 6, 7 and 8 were written by a different person. They are much easier to read. Please adjust the language in the other chapters.

Reviewer 2 Report

Here, the authors present a comprehensive and well-written review of the multiple roles of PRMT5 (alone and in cooperation with other PRMTs) in regulating gene and protein expression. The actions of PRMT5 most relevant to PDAC growth and potential inhibition strategies are effectively discussed. These topics are accompanied by detailed and nicely designed figures on key roles of PRMT5. 

Is the 5 year overall survival of 17% for all PDAC patients, or only those eligible for surgery/immunotherapy? Please add a citation.

In Figure 1, it looks like regulators of PRMT5 are listed in blue and effects of PRMT5 are in green. Consider adding this information to the figure legend.

In Figure 2, it would look a bit cleaner to be more consistent with capitalization of labels/terms used. Also, please list the three axes of PRMT5 mechanisms of acton in the figure legend.

Author Response

Here, the authors present a comprehensive and well-written review of the multiple roles of PRMT5 (alone and in cooperation with other PRMTs) in regulating gene and protein expression. The actions of PRMT5 most relevant to PDAC growth and potential inhibition strategies are effectively discussed. These topics are accompanied by detailed and nicely designed figures on key roles of PRMT5. 

Is the 5 year overall survival of 17% for all PDAC patients, or only those eligible for surgery/immunotherapy? Please add a citation.

This is 5 year OS is actually 15% based on patient diagnosed with PDAC who had underwent resection and reference included. (line 45). The original reference I could not find easily so updated figure accordingly. Thank you for pointing out the discrepancy. 

Latenstein, A.E.J.; Van Roessel, S.; Van Der Geest, L.G.M.; Bonsing, B.A.; Dejong, C.H.C.; Groot Koerkamp, B.; De Hingh, I.H.J.T.; Homs, M.Y.V.; Klaase, J.M.; Lemmens, V.; et al. Conditional Survival After Resection for Pancreatic Cancer: A Population-Based Study and Prediction Model. Annals of Surgical Oncology 2020, 27, 2516-2524, doi:10.1245/s10434-020-08235-w.

In Figure 1, it looks like regulators of PRMT5 are listed in blue and effects of PRMT5 are in green. Consider adding this information to the figure legend.

Updated in figure legend as suggested.

In Figure 2, it would look a bit cleaner to be more consistent with capitalization of labels/terms used. Also, please list the three axes of PRMT5 mechanisms of acton in the figure legend.

Updated the terms under CD8+ cells

Make alternative splicing font smaller in TME

Added the color legend and also to the figure
